# RNA polymerase III is involved in regulating *Plasmodium falciparum* virulence

Gretchen Diffendall[1,2], Aurelie Claes[1], Anna Barcons-Simon[1,2,3], Prince Nyarko[4], Florent Dingli[5], Miguel M Santos[6], Damarys Loew[5], Antoine Claessens[4,7], Artur Scherf[1]*

[1]Institut Pasteur, Universite Paris Cité, Paris, France; [2]Institut Pasteur, Sorbonne Université Ecole doctorale Complexité du Vivant, Paris, France; [3]Institut Pasteur, Biomedical Center, Division of Physiological Chemistry, Faculty of Medicine, Ludwig-Maximilians-Universität München, Munich, Germany; [4]Institut Pasteur, Laboratory of Pathogen-Host Interaction (LPHI), CNRS, University of Montpellier, Montpellier, France; [5]Institut Pasteur, Institut Curie, PSL Research University, Centre de Recherche, CurieCoreTech Mass Spectrometry Proteomics, Paris, France; [6]Institut Pasteur, Instituto de Medicina Molecular João Lobo Antunes, Faculdade de Medicina da Universidade de Lisboa, Lisboa, Portugal; [7]Institut Pasteur, LPHI, MIVEGEC, CNRS, INSERM, University of Montpellier, Montpellier, France

*For correspondence:
artur.scherf@pasteur.fr

Competing interest: The authors declare that no competing interests exist.

**Abstract** While often undetected and untreated, persistent seasonal asymptomatic malaria infections remain a global public health problem. Despite the presence of parasites in the peripheral blood, no symptoms develop. Disease severity is correlated with the levels of infected red blood cells (iRBCs) adhering within blood vessels. Changes in iRBC adhesion capacity have been linked to seasonal asymptomatic malaria infections, however how this is occurring is still unknown. Here, we present evidence that RNA polymerase III (RNA Pol III) transcription in *Plasmodium falciparum* is downregulated in field isolates obtained from asymptomatic individuals during the dry season. Through experiments with in vitro cultured parasites, we have uncovered an RNA Pol III-dependent mechanism that controls pathogen proliferation and expression of a major virulence factor in response to external stimuli. Our findings establish a connection between *P. falciparum* cytoadhesion and a non-coding RNA family transcribed by Pol III. Additionally, we have identified *P. falciparum* Maf1 as a pivotal regulator of Pol III transcription, both for maintaining cellular homeostasis and for responding adaptively to external signals. These results introduce a novel perspective that contributes to our understanding of *P. falciparum* virulence. Furthermore, they establish a connection between this regulatory process and the occurrence of seasonal asymptomatic malaria infections.

## eLife assessment

This **important** study links the activity of polymerase III to the regulation of virulence gene expression in the deadliest malaria parasite, *Plasmodium falciparum*. It identifies Maf1 as a Pol III inhibitor that enables the parasite to respond to external stimuli such as magnesium chloride plasma levels by downregulating Pol III-transcribed ruf6 genes and subsequently regulated var genes. While the evidence presented is generally **convincing**, some of the results are **incomplete**, and the mechanistic link between external signals and Maf1 activation remains unknown.

## Introduction

The parasite *Plasmodium falciparum* is responsible for the deadliest form of human malaria that annually affects over 200 million people, with 619,000 fatal cases, majorly African children under the age of 5. The disease is transmitted to its human host during a blood meal by the parasites' vector, *Anopheles* mosquitoes. Disease symptoms are seen while the parasite multiplies asexually within the host RBCs. During this time, variant surface antigens (VSAs) are exported and displayed on the surface of the infected red blood cells (iRBCs) (*Wahlgren et al., 2017*). During the ~48 hr asexual intraerythrocytic developmental cycle, the parasite develops through different morphological stages: ring (circulating), trophozoite, and schizont (sequestered in capillaries). Expression of one type of VSA that is linked to immune evasion and pathogenesis, termed *P. falciparum* erythrocyte membrane protein 1 (PfEMP1) (*Leech et al., 1984*), can bind to a wide range of receptors on endothelial cells such as CD36. PfEMP1 binding mediates adhesion to the vascular endothelium within the host, thereby preventing mature iRBCs from traveling to, and being cleared by, the spleen (*Smith et al., 2013*). *Plasmodium* parasites replicate via schizogony inside the host RBCs generating the infectious form, termed merozoites. Upon bursting of the iRBC, merozoites invade new RBCs, thus enabling the cycle to restart. The number of merozoites per schizont has been shown to reduce in unfavorable conditions as a way to diminish the total parasite load and therefore overall disease severity (*Mancio-Silva et al., 2017*).

*P. falciparum* chronic infection relies largely on mutually exclusive expression of PfEMP1 surface adhesins, that are encoded by the *var* virulence gene family (*Scherf et al., 1998*; *Smith et al., 1995*; *Guizetti and Scherf, 2013*). The expression of *var* genes, while under tight epigenetic regulation by the parasite, is additionally controlled by a family of ncRNA, termed RUF6. *P. falciparum* 3D7 encodes 15 RUF6 genes dispersed over several chromosomes located adjacent to central *var* genes (*Gardner et al., 2002*). RUF6 ncRNA have been observed to associate with the active *var* gene in trans and transcriptional repression disrupts the monoallelic expression of *var* genes by downregulating the entire *var* gene family (*Barcons-Simon et al., 2020*; *Guizetti et al., 2016*). Genetically modified parasites that express the entire RUF6 gene family show a general upregulation of *var* genes (*Zhang et al., 2014*; *Fan et al., 2020*).

*P. falciparum* virulence is linked to cytoadhesion and sequestration as a result of reduced clearance of iRBCs in the spleen and increased microcapillary obstruction and local inflammation (*Miller et al., 2002*). Substantial sequestration of iRBCs in critical target organs has been associated with severe malaria (*Miller et al., 2013*; *Guillochon et al., 2022*). In long-term asymptomatic infections, found during prolonged low transmission dry seasons in malaria-endemic regions, decreased cytoadhesion and longer circulation time of iRBCs in the blood has been observed (*Andrade et al., 2020*). It is not fully understood how asymptomatic infections develop and are maintained nor is a molecular understanding of host and parasite factors, that underlies disease severity during the dry and wet seasons (reviewed in *Zhang and Deitsch, 2022*). Disease severity appears to be influenced by a variety of factors derived from the human host, parasite, and environment. *Plasmodium* parasites are auxotrophic for various nutrients including glucose, certain types of amino acids, and lipids, which must be obtained from the host. Physiological conditions in the host, like diabetes, pregnancy, diet, and immunity, can change the metabolic environment the parasites are exposed to. Numerous studies conclude that host immunity is a contributing factor to the maintenance of asymptomatic infections (*Kimenyi et al., 2019*). While it is proposed that *P. falciparum* virulence regulation contributes to decreased endothelial binding, the mechanism and factors involved remains to be determined.

Almost all eukaryotes use conserved nutrient sensing pathways, including mTOR, GCN2, GCN5, Rgt2/Snf3, and Snf1/AMPK signaling cascades (reviewed in *Kumar et al., 2021*). Most notably, the target of rapamycin complex (TORC) pathway involves the signaling of external factors to control many cellular processes (*Wullschleger et al., 2006*). Specifically, TORC participates in the regulation of RNA polymerase III (RNA Pol III) activity via the phosphorylation of its inhibitor, Maf1. When nutrients are available and mTOR kinase is active, Maf1 is hyperphosphorylated resulting in an inactive cytoplasmic state. Stress-induced Maf1 dephosphorylation results in nuclear localization and inhibition of RNA Pol III (*Pluta et al., 2001*; *Rollins et al., 2007*; *Vannini et al., 2010*). This pathway responds to a range of positive and negative external stimuli, most notably the presence of amino acids, that drive or inhibit cellular growth. The majority of the TORC pathway components are lost in the apicomplexan lineage (*McLean and Jacobs-Lorena, 2017*; *Serfontein et al., 2010*; *van Dam et al., 2011*). While

*P. falciparum* encodes none of the core TORC1 components, it does contain a homologue to Maf1 (*McLean and Jacobs-Lorena, 2017*). The biology of *P. falciparum* Pf-Maf1 is still poorly characterized.

Here, we present data showing inhibition of RNA Pol III transcription in field isolates from asymptomatic individuals during the dry season. We present evidence for *P. falciparum* Maf1 as a negative regulator of RNA Pol III-transcribed genes including tRNA and RUF6 ncRNA, and demonstrate that this mechanism is responsive to external stimuli using in vitro culture. Inhibition of *P. falciparum* cytoadherence, via the regulation of RNA Pol III activity, represents a new paradigm contributing to *P. falciparum* virulence and links this process to seasonal asymptomatic malaria infections.

## Results

### RUF6 ncRNA is downregulated in asymptomatic individuals during the dry season

We and others have previously established the role of the 15-member RUF6 ncRNA gene family in *P. falciparum var* gene expression (*Barcons-Simon et al., 2020*; *Diffendall et al., 2023*; *Fan et al., 2020*; *Guizetti et al., 2016*). RUF6 ncRNA protein complex associates with the active *var* gene expression site (*Diffendall et al., 2023*; *Guizetti et al., 2016*), and knockdown (KD) of the ncRNA RUF6 gene family dramatically reduced *var* gene transcription (*Barcons-Simon et al., 2020*). Bioinformatic analysis identified highly conserved RNA Pol III binding elements (conserved A- and B-box sequence) within RUF6 genes (*Guizetti et al., 2016*; *Figure 1—figure supplement 1A*), hinting at a role for RNA Pol III in virulence gene regulation. Thus, we determined the effects of RNA Pol III inhibition on gene transcription in synchronized wildtype parasites. We observed downregulation of tRNA and RUF6 transcription after 21 hr of treatment with a commercially available RNA Pol III inhibitor (*Figure 1—figure supplement 1B*). As expected, no changes were observed in the transcription of an RNA Pol II-regulated gene, *uce* (ubiquitin-conjugating enzyme, PF3D7_0812600; *Figure 1—figure supplement 1B*). As predicted from the RUF6 sequence, our data provide direct experimental evidence for RNA Pol III-dependent transcription of RUF6 genes.

We next wanted to determine if RNA Pol III-transcribed genes, specifically RUF6 ncRNA, varied between parasites causing different clinical states of malaria infection, since previous studies did not include RNA Pol III-transcribed genes for their data analysis (*Andrade et al., 2020*). We first re-analyzed raw RNA-seq data, from a previous study that compared parasites infecting individuals during the dry versus wet season in a malaria-endemic region of Mali (*Andrade et al., 2020*), to include these genes. We were able to retrieve transcriptional data for two RNA Pol III-transcribed tRNAs (tRNA Leucine Pf3D7_API05400 and tRNA Asparagine Pf3D7_API05500, *Figure 1—figure supplement 2*), both of which show significantly lower levels in parasites infecting asymptomatic individuals during the dry season (n=12) when compared to parasites infecting symptomatic individuals during the wet season (n=12).

We further investigated whether RNA Pol III modulates gene transcription in field isolates of *P. falciparum* from different individuals infected during the wet and dry season in malaria-endemic regions (The Gambia). In this pilot study, we analyzed *P. falciparum* steady-state levels of RNA in venous blood samples taken from individuals with asymptomatic parasitemia during the dry season (n=17), as well as mildly symptomatic malaria infections during the wet season (n=14) in The Gambia (*Collins et al., 2022*; *Fogang et al., 2024*). Reverse transcription-quantitative PCR (RT-qPCR) of total RNA showed that levels of tRNAs (Asparagine and Valine) were significantly lower (p<0.05) in parasites from asymptomatic infections during the dry season compared to parasites causing symptomatic infections during the wet season (*Figure 1A and B*, *Figure 1—source data 1*). Likewise, RUF6 ncRNA was significantly downregulated (p<0.005) in parasites from asymptomatic infections during the dry season when compared to symptomatic infections during the wet season (*Figure 1C*). Expression was normalized to a housekeeping gene encoding fructose bisphosphate aldolase (Pf3D7_1444800).

The same previous study, mentioned above, also showed that *var* gene transcription varied in parasites infecting individuals during the dry versus wet seasons (*Andrade et al., 2020*). We observed a significantly lower level of *var* gene transcription (p<0.005) in parasites taken from asymptomatic individuals during the dry season compared to those taken from symptomatic individuals during the wet season (*Figure 1D*). The observed lower transcript levels of *var* genes may be a direct result of reduced RUF6 ncRNA levels. Taken together, our results reveal that levels of RNA Pol III-transcribed

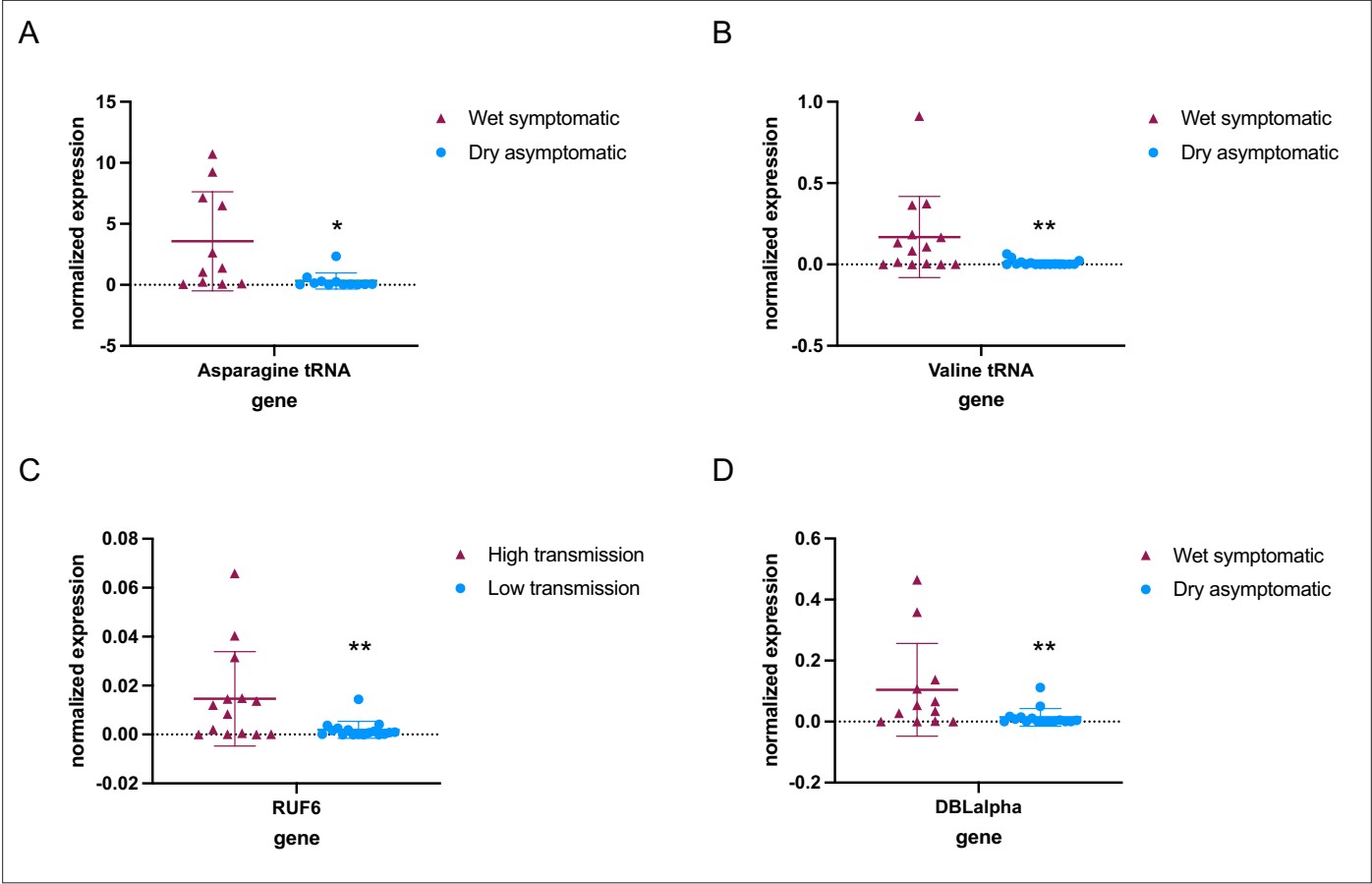

**Figure 1.** RNA polymerase III (RNA Pol III)-transcribed genes are downregulated in asymptomatic individuals during the dry season. Steady-state RNA levels as quantified by reverse transcription-quantitative PCR (RT-qPCR) using primers to tRNA Asparagine (Pf3D7_0714700) (**A**) and tRNA Valine (Pf3D7_0312600) (**B**) as well as RNA Pol III-transcribed RUF6 ncRNA (**C**). DBLalpha primers were used to detect RNA Pol II-transcribed *var* genes (**D**). Normalized expression is shown using fructose-bisphosphate aldolase (FBA Pf3D7_1444800) as the reference gene in symptomatic individuals during the wet season (n = 14[+]) and asymptomatic individuals during the dry season (n = 17[+]). [+] with the exception of Asparagine tRNA wet symptomatic (n=11) and dry asymptomatic (n=12), and DBLalpha wet symptomatic (n=12) and dry asymptomatic (n=16). Boxplots indicate the mean with standard deviation. Wilcoxon matched-pairs signed rank test was done to determine significance (* indicates p<0.05 and ** indicates p<0.005).

The online version of this article includes the following source data and figure supplement(s) for figure 1:

**Source data 1.** Reverse transcription-quantitative PCR (RT-qPCR) values used in analysis.

**Figure supplement 1.** RUF6 is transcribed by RNA polymerase III (RNA Pol III).

**Figure supplement 2.** RNA polymerase III (RNA Pol III)-transcribed genes are reduced during dry asymptomatic cases.

genes, particularly the regulatory RUF6 ncRNA, varies significantly between two different clinical states and seasons.

## The environmental factor, magnesium, plays a regulatory role in genes transcribed by RNA Pol III

In studies of malaria infections, variations in metabolite levels were observed between symptomatic and asymptomatic cases across wet and dry seasons. However, the data did not pinpoint any particular factor that could be reliably linked to a specific condition of the disease (*Andrade et al., 2020*). Other clinical studies investigated the correlation between macro- and micro-mineral concentrations and disease severity. In one study, it was discovered that levels of serum $MgCl_2$ were lower in cases of more severe malaria infections (*Innocent et al., 2013*). We chose to analyze plasma samples from individuals participating in the same study as referenced in *Figure 1*, focusing on magnesium concentrations. Our findings show a significant increase in $MgCl_2$ levels in asymptomatic individuals during the dry season compared to symptomatic individuals in the wet season, as depicted in *Figure 2A*.

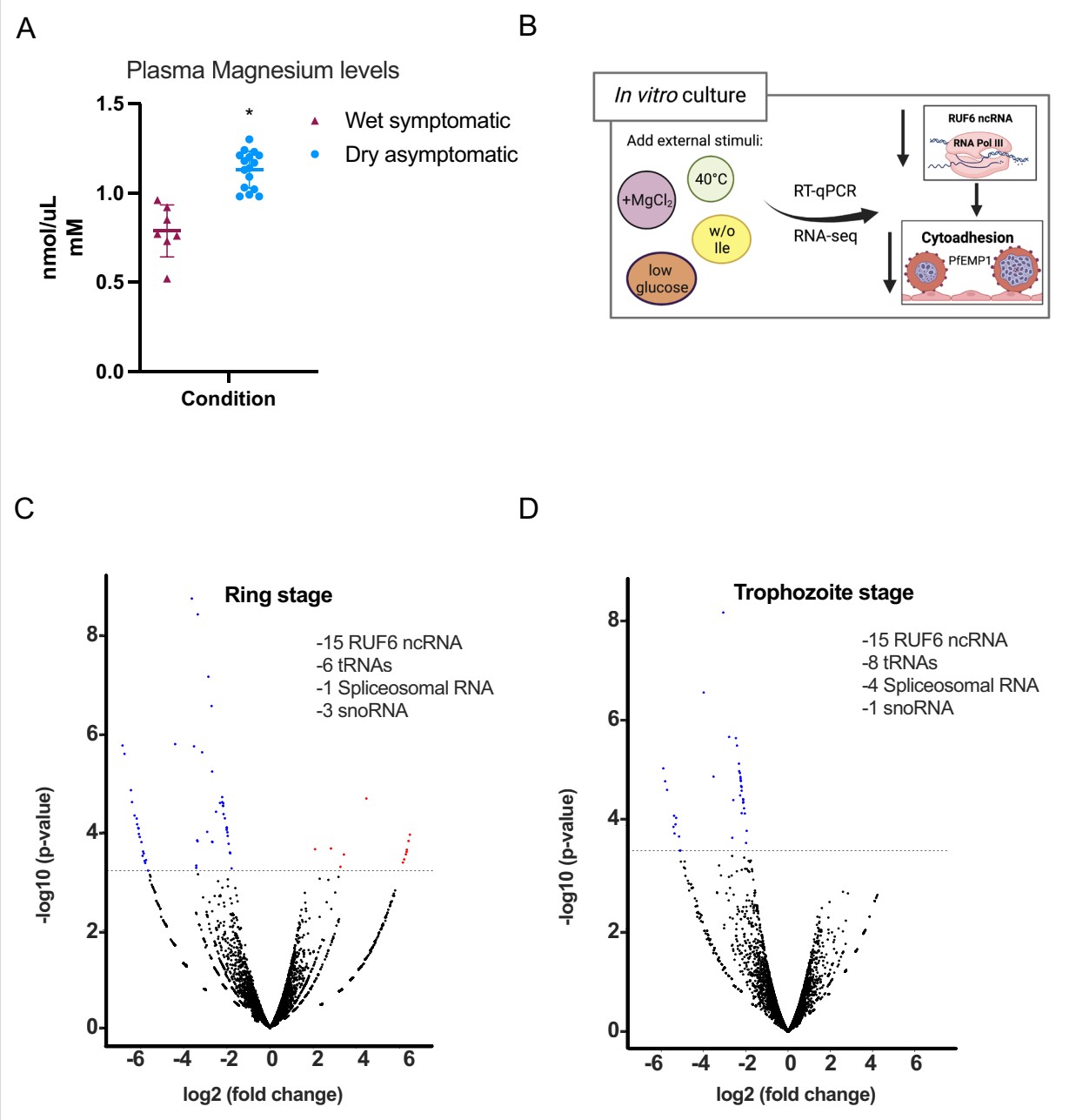

**Figure 2.** External factors modulate RNA polymerase III (RNA Pol III)-transcribed genes. (**A**) Plasma magnesium levels are significantly increased in asymptomatic individuals during the dry season compared to symptomatic individuals during the wet season. Concentration is shown in nmol/µL (mM). (**B**) Schematic showing underlying molecular mechanism summary using in vitro cultured *P. falciparum*. (**C, D**) Volcano plot showing $\log_2$(fold-change, FC) against $-\log10$ (p-value) of transcripts identified by RNA-seq analysis of 3D7 control and addition of $MgCl_2$ harvested during ring (**C**) and trophozoite stages (**D**). Expressed transcripts from three replicates between control and addition of $MgCl_2$ that are significantly upregulated are highlighted in red while significantly downregulated RNA Pol III genes are highlighted in blue (FDR corrected p-value of <0.05 and an FC≥±1.95) with examples listed as text. Black dots indicate non-significant transcripts with an FC≤2.0.

The online version of this article includes the following source data and figure supplement(s) for figure 2:

**Source data 1.** RNA-seq analysis for control and $MgCl_2$ supplemented cultured parasites.

**Figure supplement 1.** RNA polymerase III (RNA Pol III)-transcribed genes are reduced in response to external factors.

**Figure supplement 2.** RNA polymerase III (RNA Pol III)-transcribed genes are reduced in response to increasing $MgCl_2$ supplementation concentrations.

Our subsequent aim was to delve into the potential molecular mechanisms connecting external factors, like magnesium, with the changes in RUF6 ncRNA levels as shown in *Figure 2B*. For this, we employed in vitro cultured *P. falciparum* asexual blood stage parasites. We supplemented culture medium with MgCl$_2$, noted for its varying levels in individuals with malaria (see *Figure 2A*). In addition, we mimicked nutrient deprivation with isoleucine-deprived culture medium (as was described in *Babbitt et al., 2012*; *McLean and Jacobs-Lorena, 2017*), fever conditions with incubation at 40°C, and food starvation with low glucose levels (*Jensen et al., 1983*; *Fang et al., 2004*). We evaluated the steady-state levels of RNA Pol I- (rRNA A1), RNA Pol II- (UCE and the active *var*), and RNA Pol III- (tRNA Valine and RUF6 ncRNA) transcribed genes in clonal parasites at late ring stage. We observed that high-temperature and low glucose conditions affected gene transcription of RNA Pol I, II, and Pol III (*Figure 2—figure supplement 1*). Deprivation of isoleucine and MgCl$_2$ supplementation seemingly affected only RNA Pol III-transcribed genes with the exception of the RNA Pol II-transcribed active *var* gene, Pf3D7_1240900, likely as a result of decreased RUF6 ncRNA as was reported earlier (*Barcons-Simon et al., 2020*).

To explore the underlying molecular pathway that leads to the downregulation of RNA Pol III activity in malaria parasites, we continued using MgCl$_2$ supplementation. We selected a concentration of 3 mM total MgCl$_2$, based on reports indicating it does not inhibit the growth of *P. falciparum* asexual blood stage and is within a physiological range (*Jahnen-Dechent and Ketteler, 2012*; *Hess et al., 1995*). Additionally, we verified lower magnesium concentrations as low as 1 mM total MgCl$_2$, observed in healthy individuals, impacts both Pol III-transcribed RUF6 and tRNA, while not affecting Pol II-transcribed UCE (*Figure 2—figure supplement 2A*). RNA-seq was done on control and MgCl$_2$ supplemented cultured parasites during two time points of the asexual blood stage parasites (ring and trophozoite) to further confirm that RNA Pol III is the main target (*Figure 2C and D*, *Figure 2—source data 1*). RNA-seq data was compared to the microarray time course data in *Bozdech et al., 2003*, as in *Lemieux et al., 2009*, which provided a statistical estimation of cell cycle progression at 12 and 24 hpi in both control and MgCl$_2$ supplemented parasites (*Figure 2—figure supplement 2B*).

These data indicate that any differences in transcription were due to MgCl$_2$ supplementation and not differences in cell cycle progression. During ring stage (*Figure 2C*), 77% of the downregulated genes contain typical RNA Pol III promoter sequences, A- and B-box consensus (from *Figure 1—figure supplement 1A*). A total of 27 ncRNA genes were downregulated (logFC<1.95, FDR<0.05) including all 15 RUF6 ncRNA, 6 tRNAs, 1 spliceosomal RNA, and 4 snoRNA. Additionally, 12 protein coding genes were upregulated with no conserved pathway nor molecular or biological process. During trophozoite stage (*Figure 2D*), 74% of the downregulated genes have typical RNA Pol III promoter sequences. This demonstrates that MgCl$_2$ supplementation predominantly inhibits RNA Pol III-transcribed genes, including the entire RUF6 gene family.

## Nuclear PfMaf1 protein is essential to regulate RNA Pol III

In almost all eukaryotes, the TORC pathway has been shown to participate in the regulation of RNA Pol III activity via the phosphorylation of its only known inhibitor, Maf1 (*Figure 3A*, reviewed in *Wullschleger et al., 2006*). Upon phosphatase activation, modulated in response to external stimuli such as low nutrient availability, Maf1 shuttles to the nucleus to repress RNA Pol III activity (*Michels et al., 2010*). Because Maf1 is conserved in the apicomplexan lineage, a predicted essential blood stage protein in *P. falciparum*, we set out to explore if *P. falciparum* Maf1 (PfMaf1) functions similarly to model eukaryotes. We generated a PfMaf1 parasite line tagged with a 3HA tag and a ligand-controlled destabilization domain (ddFKBP), PfMaf1-FKBP (*Figure 3B*, top). PfMaf1-FKBP transfected parasite clonal populations were validated for proper integration and confirmation that the system was working, shown by western blot for addition and removal of the ligand, Shield-1, over the course of 3 cycles (*Figure 3B*, bottom, *Figure 3—source data 1 and 2*). We observed that upon removal of Shield-1, PfMaf1 levels were shown to decrease by approximately 57% in total extracts after 1 cycle and almost complete degradation was achieved after 2 cycles.

Maf1 in *Plasmodium* species seemingly affects asexual blood stage proliferation making classical gene knockout (KO) experiments unattainable (*McLean and Jacobs-Lorena, 2017*; *Zhang et al., 2018*), pointing to a regulatory role of the putative RNA Pol III inhibitor under normal in vitro culture growth conditions. Once PfMaf1 is depleted (*Figure 3B*, bottom, 3 cycles −Shield), transcription of tRNA Valine and RUF6 ncRNA increased dramatically by threefold (*Figure 3C*), possibly perturbing

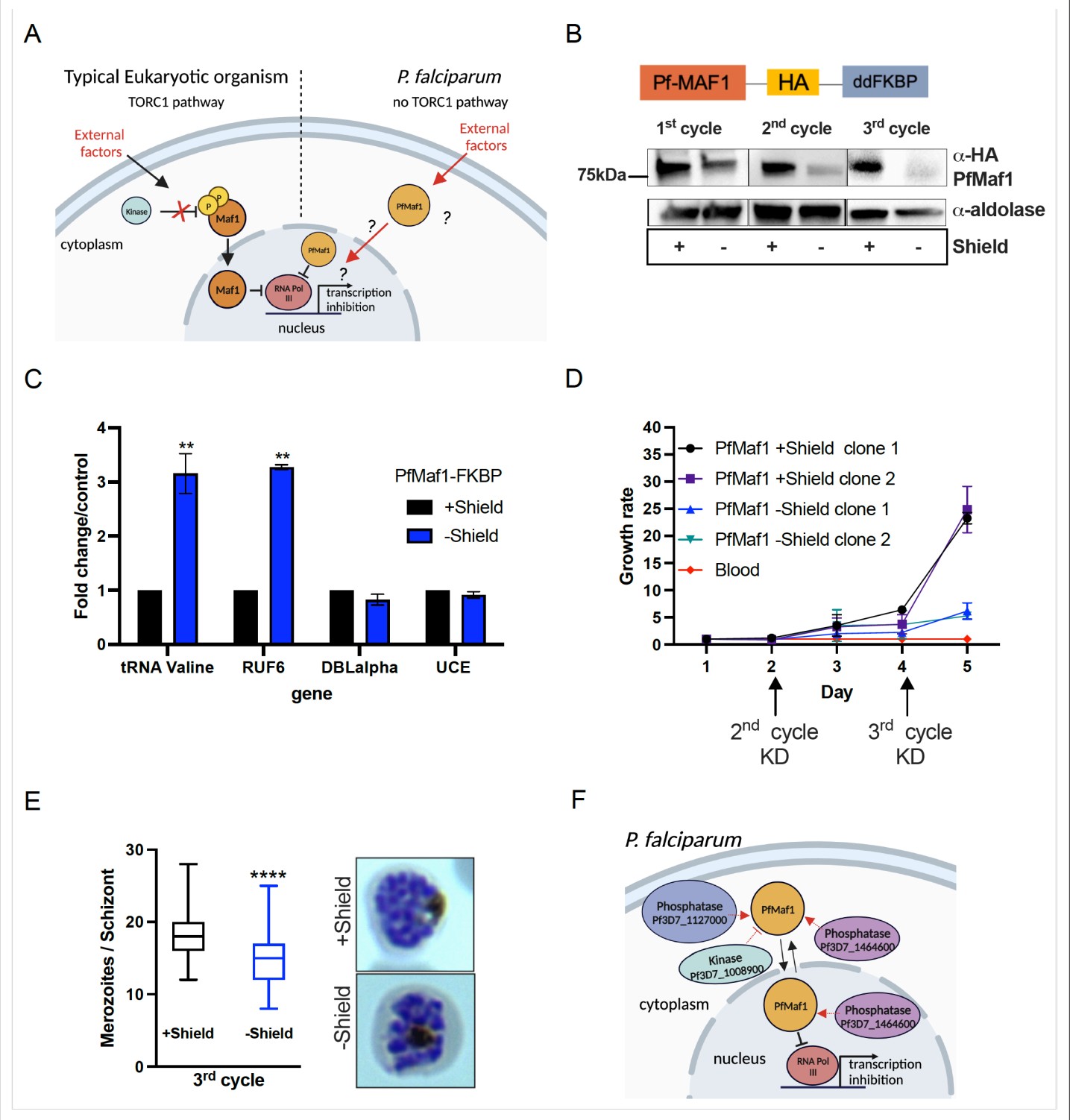

**Figure 3.** Nuclear PfMaf1 is essential to regulate RNA polymerase III (RNA Pol III). (**A**) Illustration of the TORC1-dependent cellular localization of Maf1 protein in unfavorable conditions in typical eukaryotic organisms compared to *P. falciparum* that has no TORC1 pathway. (B, top) Illustration of recombinant PfMaf1 with a 3HA tag followed by a ddFKBP domain to allow for knockdown studies. (B, bottom) Western blot analysis for PfMaf1 in total extracts in pSLI-Maf1-FKBP transfected parasites at 24 hpi after 1, 2, and 3 cycles without addition of Shield-1 (−) and control, with addition of Shield-1 (+). Aldolase levels are also shown. Representative of three replicates. (**C**) Transcript levels as quantified by reverse transcription-quantitative PCR (RT-qPCR) using the same primers in *Figure 2B* in parasites harvested at 18 hpi in control group and without Shield for 2 cycles. Error bars are displayed from three biological replicates. Statistical significance was determined by two-tailed Student's t-test (**p<0.005). (**D**) Growth curve over

*Figure 3 continued on next page*

*Figure 3 continued*

5 days of clonal pSLI-Maf1-FKBP parasites for two conditions: in the presence or absence of Shield-1. Uninfected red blood cells ('blood' in red) serve as reference of background. Error bars indicate standard deviation of three technical replicates in different blood from two different clones (n=6). (**E**) Data is represented as box-whisker plot of mean merozoite number per schizont ± SD (Mann-Whitney), with the median represented at the center line. Boxplots show the data of 100 segmented schizonts counted per condition (n=100). Statistical significance was determined by two-tailed Student's t-test (***p<0.001). Representative Giemsa images are shown to the right for + and – Shield. (**F**) Visual representation of co-immunoprecipitation followed by quantitative mass spectrometry (Co-IP MS) analysis of cytoplasmic and nuclear PfMaf1. Labeled proteins represent important significant and unique proteins in cytoplasmic and nuclear fractions not found in either of the controls.

The online version of this article includes the following source data and figure supplement(s) for figure 3:

**Source data 1.** Western blot analysis shows PfMaf1 is depleted after 1 and 3 cycles without Shield-1 drug added.

**Source data 2.** Western blot analysis shows PfMaf1 is depleted after 2 cycles without Shield-1 drug added.

**Source data 3.** Co-immunoprecipitation followed by quantitative mass spectrometry (Co-IP MS) analysis for cytoplasmic and nuclear PfMaf1-binding proteins.

**Figure supplement 1.** PfMaf1 cytoplasmic and nuclear interactome.

Co-immunoprecipitation followed by quantitative mass spectrometry (Co-IP MS) volcano plot of enrichment for all five replicates for cytoplasmic (**A**) and nuclear (**B**) PfMaf1 vs control proteins are indicated and labeled. Each dot represents a protein, and its size corresponds to the sum of peptides from both conditions used to quantify the ratio of enrichment. x-axis = log$_2$(fold-change), y-axis = −log$_{10}$(p-value), horizontal red line indicates adjusted p-value=0.05, and vertical green lines indicate absolute fold-change=2.0. Side panels indicate proteins uniquely identified in either sample (y-axis=number of peptides per 100 amino acids) with a minimum of two distinct peptides in three replicates of a same state. (**C**) Table showing values for significantly and uniquely enriched proteins from both extracts as labeled in *Figure 3F*. (**D**) Volcano plot showing the distribution of significant and unique proteins in cytoplasmic and nuclear fractions not found in either of the controls. Each dot represents a protein, and its size corresponds to the sum of peptides from both conditions used to quantify the ratio of enrichment. x-axis = log$_2$(fold-change), y-axis = −log$_{10}$(p-value), horizontal red line indicates adjusted p-value=0.05, and vertical green lines indicate absolute fold-change=2.0. Side panels indicate proteins uniquely identified in either sample (y-axis=number of peptides per 100 amino acids) with a minimum of two distinct peptides in three replicates of a same state.

cellular homeostasis. RNA Pol II-transcribed genes were unaffected. Parasite growth was assessed over the course of 5 days for two PfMaf1-FKBP clones cultured in the presence or absence of PfMaf1 (+/−Shield-1, removed 1 cycle before day 0, *Figure 3D*). A significant difference in growth was achieved, most evidently in parasites cultured without Shield for 3 cycles, consistent with transposon mutagenesis scoring PfMaf1 essential (*Zhang et al., 2018*). Dead parasites were not detected by Giemsa staining even up to 8 cycles without Shield-1. We confirmed that differences in merozoite numbers per schizont contributed to the change in growth rate between the control and KD PfMaf1 parasites after the third cycle KD (*Figure 3E*).

Additionally, we demonstrated that nuclear PfMaf1 interacts with the RNA Pol III protein complex by performing co-immunoprecipitation followed by quantitative mass spectrometry (Co-IP MS) of cytoplasmic and nuclear fractions of PfMaf1-HA-ddFKBP (*Figure 3—source data 3*). Each fraction was split to include a control with no antibody added. Analysis of the quantitative mass spectrometry data revealed unique interactome of cytoplasmic PfMaf1 and nuclear PfMaf1 in their respective samples compared to their controls (*Figure 3—figure supplement 1*). Gene Ontology (GO) analysis showed that PfMaf1-associated proteins uniquely found in the nuclear fraction, and neither of the two controls nor cytoplasmic fraction are significantly represented by the biological function category of 'RNA polymerase III transcription' (p=0.00175) with two subunits of RNA Pol III (Pf3D7_1206600 and Pf3D7_1329000). (GO) analysis of significant and unique proteins found only in the cytoplasmic fractions, and not nuclear, are significantly represented by the molecular function category of 'protein kinase activator' (p=0.0143, Pf3D7_1103100). Additionally, two protein phosphatases (Pf3D7_1127000 and Pf3D7_1464600) were significantly enriched in the cytoplasmic fraction (*Figure 3—figure supplement 1*). *Figure 3F* illustrates the cytoplasmic and nuclear enzymes associated with PfMaf1, which could play a role in sensing environmental shifts that lead to the inhibition of Pol III.

## Magnesium modulates cytoadhesion through PfMaf1-regulated RNA Pol III transcription

We further investigated if nuclear PfMaf1 activity could be regulated, in response to external factors, to lead to changes in RNA Pol III activity using *P. falciparum* wildtype parasites cultured in vitro. The *P. falciparum* cellular localization of PfMaf1 was investigated in response to external stimuli (+MgCl$_2$). Parasites were tightly synchronized, split into control, addition of MgCl$_2$, and harvested during late ring

stage. In contrast to several eukaryotic model systems, under normal culture conditions we observed PfMaf1 in cytoplasmic and nuclear fractions using immunoprecipitation followed by western blot analysis (**Figure 4A**, **Figure 4—figure supplement 1A**, **Figure 4—source data 1**). Nuclear PfMaf1 levels increased upon addition of $MgCl_2$ compared to control nuclear PfMaf1 (**Figure 4A**). Immunofluorescence (IF) assays showed that PfMaf1 forms foci-like aggregates in the cytoplasm and near the nuclear periphery in both culture conditions (**Figure 4—figure supplement 1B**).

Next, we used the PfMaf1-FKBP transfected parasite lines, to show that the downregulation of RNA Pol III-transcribed genes, triggered by external stimuli, is dependent on PfMaf1. We used the PfMaf1 KD parasites (second cycle, see **Figure 3B**) in control conditions and $MgCl_2$ supplementation to the growth medium in ring stage parasites for RT-qPCR (**Figure 4B**). We confirmed the previously observed downregulation of RNA Pol III-transcribed tRNA Valine and RUF6 upon $MgCl_2$ supplementation in two independent PfMaf1-FKBP parasite clones. Notably, when PfMaf1 degradation was induced, the addition of $MgCl_2$ did not affect Pol III transcription. These results demonstrate that PfMaf1 mediates the observed decrease in RNA Pol III-transcribed genes with $MgCl_2$ supplementation.

Previous attempts to develop a KO line did not yield success, as highlighted by McLean and Jacobs-Lorena in 2017. However, our strain with an inducible system for degrading PfMaf1 protein has unveiled a twofold function of PfMaf1: (i) to help maintain Pol III transcription at levels essential for optimal parasite proliferation, and (ii) to serve as a key element in an environmental sensing pathway that directly controls Pol III activity.

Finally, we set out to link *P. falciparum* virulence gene expression with decreased RUF6 transcription, by investigating the effect of $MgCl_2$ supplementation on cytoadherence of 3D7 iRBCs. We found that in static binding assays to the endothelial receptor CD36, bound to plastic dishes, adhesion of iRBCs was reduced by 50% with $MgCl_2$ supplementation (**Figure 4C**). This was also confirmed at the protein level using antibodies directed against the conserved intracellular ATS domain of PfEMP1 (**Figure 4D**, **Figure 4—source data 2**). Our findings, which demonstrate the inducible inhibition of cytoadhesive capacity, are particularly significant for asymptomatic *P. falciparum* infections, as illustrated in **Figure 4E**.

## Discussion

Our study reveals a regulatory mechanism in *P. falciparum* involving RNA Pol III, which plays a pivotal role in the parasite's virulence. This discovery illuminates a previously unidentified adaptive molecular process in the parasite's cytoadhesion and proliferation. We propose a connection with asymptomatic infections prevalent during the dry season in African regions, a period characterized by reduced mosquito transmission. In our analysis, we compared RNA from parasites in symptomatic individuals during the wet season to those in asymptomatic individuals in the dry season. In this comparison, we noted a reduction in the levels of tRNAs and RUF6 ncRNA, with the latter being recognized for its involvement in regulating *var* gene expression, as illustrated in **Figure 1A–C**. To date, RNA Pol III has not been acknowledged as a factor influencing the virulence of *P. falciparum* in mild or asymptomatic malaria infections. Additionally, our research revealed a link between elevated serum $MgCl_2$ levels and asymptomatic malaria infections, as shown in **Figure 2A**. The normal concentration range for $MgCl_2$ in human serum is between [0.7 and 1.0 mM], while mild hypermagnesemia is characterized by levels ranging from [2.2 to 3.5 mM]. In our pilot study, the average concentration was observed to be around [0.7 mM] during the wet season and increased to [1.1 mM] in the dry season.

To investigate the potential mechanistic link between the changes in $MgCl_2$ levels and Pol III activity, we established an in vitro culture assay for *P. falciparum* that enabled us to delve into the molecular mechanism underpinning the environment-dependent inhibition of Pol III. We experimented with various concentrations of $MgCl_2$ to assess its effect on Pol III activity. The standard culture medium we used contains [0.5 mM] of $MgCl_2$. Our observations indicated a progressive inhibition of Pol III activity in the range of [1–3 mM]. To further explore the molecular mechanisms involved, we opted to use a concentration of 3 mmol/L in our continued studies. Quantitative RT-PCR analysis showed inhibition of Pol III transcription of tRNA and the RUF6 ncRNA gene family. Subsequent RNA-seq data validated that transcriptional inhibition, induced by $MgCl_2$ supplementation, is primarily limited to canonical RNA Pol III-transcribed genes, which include A- and B-box containing sequences, and notably the RUF6 gene, crucial for the activation of *var* genes. Our findings established that the inhibition of *var* gene expression, mediated by Pol III, led to a decrease in the cytoadherence of iRBCs

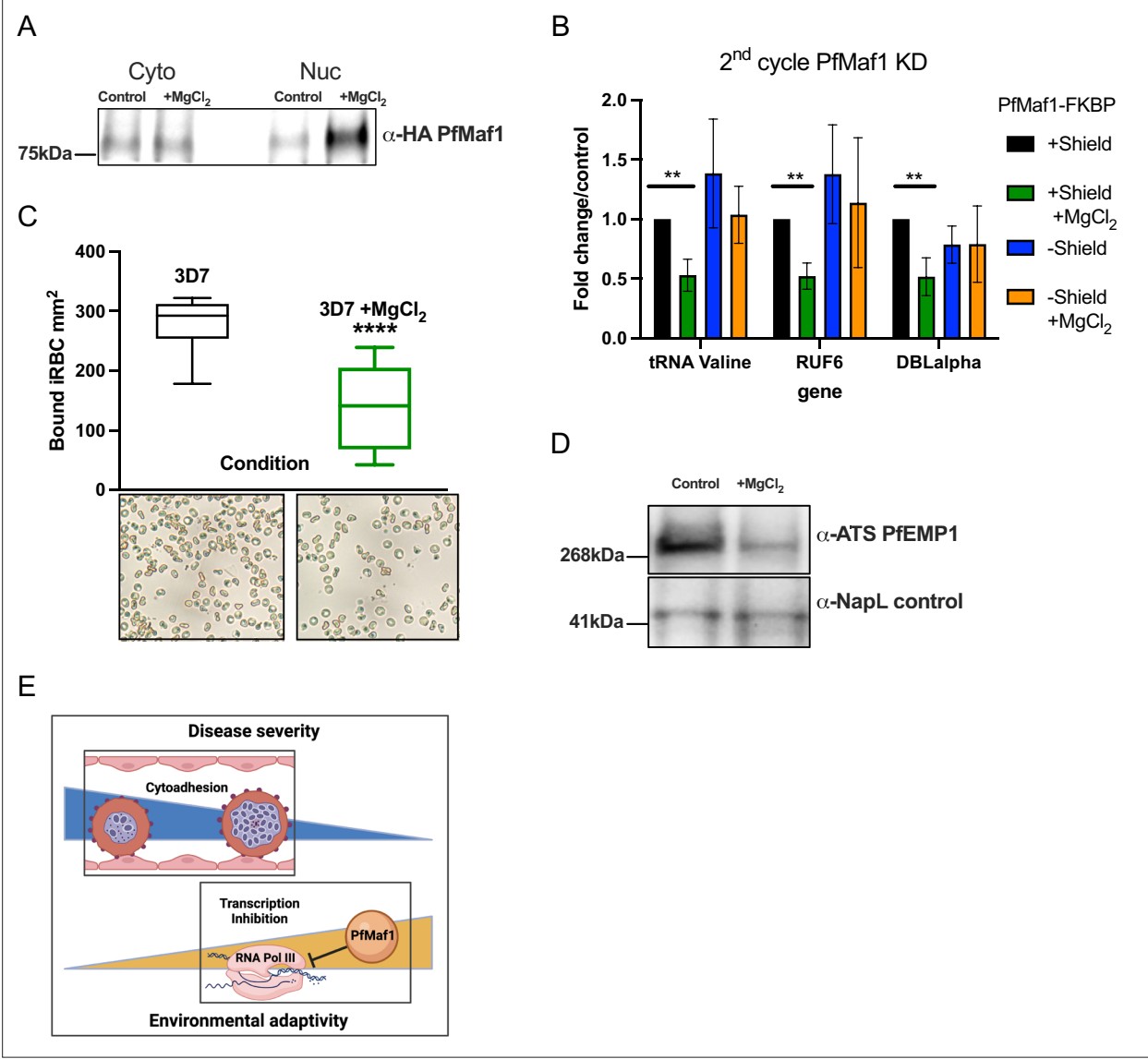

**Figure 4.** External factors modulate virulence through PfMaf1-regulated RNA polymerase III (RNA Pol III) inhibition. (**A**) Immunoprecipitation western blot analysis for cytoplasmic and nuclear extracts for PfMaf1 expression in PfMaf1-FKBP transfected parasites with addition of $MgCl_2$ ([3 mM] total) and control parasites harvested at 18 hpi. Controls are shown in **B**. (**B**) Transcript levels as quantified by reverse transcription-quantitative PCR (RT-qPCR) using primers to tRNA Valine (Pf3D7_0312600), RUF6 ncRNA, *var* DBLalpha, and normalized to FBA (Pf3D7_1444800) for four conditions: control (black), addition of $MgCl_2$ (green), knockdown (KD) of PfMaf1 (blue), and KD of PfMaf1 and addition of $MgCl_2$ (orange). Error bars are displayed from three biological replicates. Statistical significance was determined by two-tailed Student's t-test (**$p<0.005$). (**C**) Cytoadhesion binding assay data is represented as box-whisker plot of mean number of bound infected red blood cell (iRBC) ± SD (Mann-Whitney) $mm^2$, with the median represented at the center line. Boxplots show the data of three biological replicates (n=3). Statistical significance was determined by two-tailed Student's t-test (****$p<0.001$). Representative images are shown below for 3D7 and 3D7 +$MgCl_2$. (**D**) Western blot analysis for extracts for ATS-PfEMP1 expression in 3D7 control parasites and with addition of $MgCl_2$ ([3 mM] total) harvested after plasmin. NapL control levels are also shown. Representative of three replicates. (**E**) Schematic showing summary of study linking decreased cytoadherence, associated with disease severity, with increased RNA Pol III inhibition, triggered in response to external factors.

The online version of this article includes the following source data and figure supplement(s) for figure 4:

**Source data 1.** Immunoprecipitation followed by western blot analysis shows nuclear PfMaf1 levels increase with $MgCl_2$ supplementation.

**Source data 2.** Western blot analysis shows decreased PfEMP1 expression with $MgCl_2$ supplementation.

**Figure supplement 1.** $MgCl_2$ supplementation increases nuclear PfMaf1 levels.

(A) Immunoprecipitation western blot analysis for cytoplasmic and nuclear extracts for PfMaf1 expression in PfMaf1-FKBP transfected parasites with addition of $MgCl_2$ ([3 mM] total) and control parasites harvested at 18 hpi from ***Figure 4A***. Anti-HA PfMaf1, aldolase, and histone H3 are shown from

*Figure 4 continued on next page*

Figure 4 continued

input and flow-through (FT). (B) Representative immunofluorescence images show brightfield, Dapi, anti-HA PfMaf1, and Dapi-HA merge for PfMaf1 in control and addition of MgCl$_2$ ([3 mM] total) in parasites harvested and fixed at late ring stage.

when supplemented with MgCl$_2$, as shown in *Figure 4C*. We noted a reduction of over 50% in binding to the endothelial receptor CD36 in a static binding assay. It is expected that the inhibition of iRBC cytoadherence would be even more pronounced under the physiological shear stress typically found in human capillaries (*Crabb et al., 1997*).

Having established the role of Pol III in the expression of virulence genes, we subsequently showed a link between environmental factors and Maf1, which is the only known eukaryotic repressor of Pol III. We created an inducible system designed to specifically reduce PfMaf1 protein levels, enabling us to explore its influence on Pol III activity. Proteomic analysis through mass spectrometry of PfMaf1's interactome showed its association with various nuclear RNA Pol III subunits, reinforcing its function as a repressor of Pol III in *P. falciparum*. PfMaf1 showed nuclear presence under standard growth conditions. Nuclear shuttling usually increases under specific conditions (stress, nutrient starvation) by changing the phosphorylation state of the Maf1 protein in eukaryotic organisms (*Michels et al., 2010*). Our study supports a model in which a critical baseline amount of PfMaf1 in the nucleus is essential to modulate the expression of tRNAs by RNA Pol III during the multinucleated blood stage of the parasite. This regulation is essential for the parasite to balance its energy resources efficiently while producing large numbers of new infective forms, known as merozoites. Under optimal growth conditions, a single parasite undergoes several rounds of genome replication, forming approximately 30 nuclei within a shared cytoplasm. When PfMaf1 is completely removed, the transcription levels of tRNA Valine and RUF6 ncRNA increase more than threefold, while the transcription level of a Pol II-mediated housekeeping gene UCE remains unchanged. This significant elevation of tRNA transcription beyond physiological levels could potentially lead to an energy deficit within the cell, resulting in the observed reduction in the parasite's multiplication rate, as demonstrated in *Figure 3C and E*. In clinical isolates, parasite densities have been associated with malaria pathogenesis (*Nyarko and Claessens, 2021*; *Thomson-Luque et al., 2021*) and low parasite densities are also a feature of asymptomatic malaria (*Murray et al., 2017*).

Nutrient availability in the host has been demonstrated to have a profound effect on the replication of parasites using *Plasmodium berghei* as a rodent malaria model. Reports from cultured blood stage *P. falciparum* parasites showed that amino acid deficiency can alter parasite growth and survival to stress (*McLean and Jacobs-Lorena, 2017*; *Marreiros et al., 2023*). Furthermore, a recent report suggested that nutrients involved in *S*-adenosylmethionine metabolism can also affect *var* gene switching in cultured parasites (*Schneider et al., 2023*). Individuals infected with malaria, compared to healthy individuals, were found to have decreased levels (up to 54%) of plasma-free amino acids, which could be the result of a nutrient-reduced diet (*Leopold et al., 2019*). In fact, dietary intake, including energy, protein, iron, zinc, calcium, and folate, was found to decrease significantly in individuals during the dry season in areas of Africa (*M'Kaibi et al., 2015*). Blood serum magnesium levels can fluctuate in humans and were found to decrease with increasing disease severity (*Innocent et al., 2013*). In our study, plasma magnesium levels were significantly increased in asymptomatic individuals during the dry season compared to symptomatic individuals during the wet season (*Figure 2A*). Wet season samples were within the range of normal serum magnesium levels, whereas dry season samples were higher. It is noteworthy that intracellular magnesium was found to increase in yeast as a result of calorie restriction, defined as a decrease in dietary glucose intake (*Abraham et al., 2016*), indicating that our MgCl$_2$ supplementation could be mimicking low glucose conditions and thus, low nutrient availability. Although interactions between human malaria and malnutrition have been studied for many years, the evidence linking an effect between the two remains inconclusive at the mechanistic level. Our study sets the stage for exploring whether there are additional external stimuli, beyond magnesium chloride, that could activate the regulatory pathway of RNA Pol III. Given the lack of a conventional TOR pathway in malaria parasites (*Serfontein et al., 2010*; *van Dam et al., 2011*), the exact signaling pathway that activates PfMaf1 remains unknown. Our mass spectrometry analysis of PfMaf1's cytoplasmic binding partners has identified interactions with enzymes, such as phosphatases and kinases, which could potentially influence Maf1 activity. This discovery opens up new avenues for future research into the environmental sensing mechanisms that function upstream of PfMaf1.

A traditional view posits that the decreased cytoadherence seen in dry season parasites may be affected by host immunity, which impairs the parasites' adhesion capabilities. Although experimental evidence supporting this theory is missing, our research introduces a novel perspective to this critical topic by highlighting the role of metabolic changes in modulating virulence gene expression.

In summary, our research demonstrates that RNA Pol III activity, regulated by environmental factors, plays a crucial role in the proliferation of parasites and in reducing the cytoadhesive capacity of *P. falciparum*. This discovery unveils a previously unknown molecular process, significantly enhancing our understanding of subclinical parasite persistence. The insights gained from this study could pave the way for novel strategies aimed at preventing severe malaria by promoting reduced pathogen virulence.

## Materials and methods

### Parasite and serum samples from malaria infected patients

Venous blood draw of different infected individuals in The Gambia during the dry and wet seasons was collected as previously described in *Collins et al., 2022*; *Fogang et al., 2024*. The study protocol was reviewed and approved by the Gambia Government/MRC Joint Ethics Committee (SCC 1476, SCC 1318, L2015.50) and by the London School of Hygiene & Tropical Medicine ethics committee (Ref 10982). The field studies were also approved by local administrative representatives, the village chiefs. Written informed consent was obtained from participants over 18 years of age and from parents/guardians for participants under 18 years. Written assent was obtained from all individuals aged 12–17 years. The dry season months include January, February, March, April, and May. The wet season months include November and October. For additional information about each sample, see *Supplementary file 2*.

### Polymerase III inhibition assay

Parasites were treated with 50 µM of RNA Pol III inhibitor CAS 577784-91-9 (Calbiochem, Merck) after sorbitol treatment at 3±3 hpi and RNA was harvested at 24 hpi in parallel with untreated and control samples. The control was treated with the same volume of DMSO added to the inhibitor-treated flasks of stock solution.

### Clinical isolate RT-qPCR

Total RNA was extracted from TRizol using an miRNeasy minikit and performing on-column DNase treatment (QIAGEN) and continued as previously described in *Collins et al., 2022*. Transcript levels were shown by using the following primers: RUF6, Valine tRNA, Asparagine tRNA, and *var* DBLalpha were normalized to the reference gene, fructose-bisphosphate aldolase (PF3D7_1444800, *Figure 2A*, *Supplementary file 1*). The starting quantity means from three replicates were extrapolated from a standard curve of serial dilutions of 3D7 genomic DNA.

### Plasma magnesium concentration assay

Plasma magnesium levels were determined using a commercial magnesium assay kit (Sigma-Aldrich MAK026) from individuals from the same study as in *Collins et al., 2022*; *Fogang et al., 2024*. Complete information on tested individual samples can be found in *Supplementary file 2*. The protocol was followed as described in the kit and read on a Synergy 2 microplate reader for spectro-photometric reading.

### Parasite culture and synchronization

Asexual blood stage 3D7 *P. falciparum* parasites were cultured as previously described in *Lopez-Rubio et al., 2009*. Parasites were cultured in human RBCs (obtained from the Etablissement Français du Sang with approval number HS 2019-24803) in RPMI-1640 culture medium (Thermo Fisher 11875) supplemented with 10% vol/vol Albumax I (Thermo Fisher 11020), hypoxanthine (0.1 mM final concentration, C.C.Pro Z-41-M), and 10 mg gentamicin (Sigma G1397) at 4% hematocrit and under 5% $O_2$, 3% $CO_2$ at 37°C. Static parasite development was monitored by Giemsa staining. Parasites were synchronized by sorbitol (5%, Sigma S6021) lysis at ring stage, plasmagel (Plasmion, Fresenius Kabi) enrichment of late stages 24 hr later, and an additional sorbitol lysis 3 hr after plasmagel enrichment.

The 0 hr time point was considered to be 1.5 hr after plasmagel enrichment. Parasites were harvested at 1–5% parasitemia.

## External stimuli induction

Synchronized parasites were divided into control, addition of magnesium chloride ($MgCl_2$), isoleucine-deficient, low glucose, and 40°C, and harvested at 18 hpi. Parasites exposed to an addition of $MgCl_2$ were supplemented with 0.5 mM, 1.5 mM, and 2.5 mM $MgCl_2$, for a final concentration of [1 mM], [1.5 mM], [2 mM], and [3 mM] including [0.5 mM] found in RPMI. Isoleucine-deficient medium consisted of 10.3 g/L RPMI 1640 isoleucine (Ile) Drop-out medium (United States BioLogicals; catalog no. R9014), supplemented with 2.0 g/L $NaHCO_3$, 6.0 g/L HEPES, 10% vol/vol Albumax I (Thermo Fisher 11020), hypoxanthine (0.1 mM final concentration, C.C.Pro Z-41-M), and 10 mg gentamicin (Sigma G1397). Low glucose (0.25 mg/mL) RPMI was made by adding glucose (Dextrose) to glucose-free media: 2.979 g HEPES+50 mL albumax+10 mL hypoxanthine+200 µL gentamycin to final volume of 500 mL with glucose-free RPMI (11879). The pH was adjusted to 7.3 with NaOH and filter-sterilized. 40°C samples were incubated in an adjacent incubator set to 40°C under 5% $O_2$, 3% $CO_2$. Parasites were then harvested with 0.075% saponin lysis at ~2–5% parasitemia for RNA, genomic DNA, and protein extraction at 18 hpi.

## RNA isolation and RT-qPCR

RNA was isolated from synchronized parasite cultures harvested at 18 hpi after saponin lysis in 0.075% saponin in PBS, followed by one wash in Dulbecco's phosphate-buffered saline (DPBS, Thermo Fisher 14190) and resuspension in the QIAzol reagent. Total RNA was extracted using an miRNeasy minikit and performing on-column DNase treatment (QIAGEN). Reverse transcription from total RNA was achieved using SuperScript VILO (Thermo Fisher Scientific) and random hexamer primers. cDNA levels were quantified by quantitative PCR in the CFX384 real-time PCR detection system (Bio-Rad) using Power SYBR Green PCR Master Mix (Applied Biosystems) and primers from a previous study (*Mancio-Silva et al., 2010*; *Guizetti et al., 2016*). Starting quantity means of three replicates were extrapolated from a standard curve of serial dilutions of genomic DNA. Transcript levels were shown by using the following primers: RUF6, Valine tRNA, Alanine tRNA, Asparagine tRNA, *var* 58 (PF3D7_1240900), and *var* DBLalpha were normalized to the reference gene, fructose-bisphosphate aldolase (PF3D7_1444800).

## Stranded RNA-seq and analysis

Infected RBCs containing synchronized (12 and 24 hpi) parasites were lysed in 0.075% saponin (Sigma S7900) in DPBS at 37°C. The parasite cell pellet was washed once with DPBS and then resuspended in 700 µL QIAzol reagent (QIAGEN 79306). Total RNA was subjected to rRNA depletion to ensure ncRNA and mRNA capture using the RiboCop rRNA Depletion Kit (Lexogen) prior to strand-specific RNA-seq library preparation using the TruSeq Stranded RNA LT Kit (Illumina) with the KAPA HiFi polymerase (Kapa Biosystems) for the PCR amplification. Multiplexed libraries were subjected to 150 bp paired-end sequencing on a NextSeq 500 platform (Illumina). Sequenced reads (150 bp paired end) were mapped to the *P. falciparum* genome (*Gardner et al., 2002*; plasmoDB.org, version 3, release 57) using 'bwa mem' (*Li and Durbin, 2009*) allowing a read to align only once to the reference genome (option '–c 1'). Alignments were subsequently filtered for duplicates and a mapping quality ≥20 using samtools (*Li and Durbin, 2009*). Three biological replicates for $-MgCl_2$ and $+MgCl_2$ samples were analyzed for both time points.

## Generation of PfMaf1 strains

All cloning was performed using KAPA HiFi DNA Polymerase (Roche 07958846001), In-Fusion HD Cloning Kit (Clontech 639649), and XL10-Gold Ultracompetent *Escherichia coli* (Agilent Technologies 200315). Transgenic pSLI parasites were generated as previously described in *Birnbaum et al., 2017*, with the following modifications: GFP was replaced with a 3HA tag and a ddFKBP domain was added after the protein of interest, PfMaf1. For localization and KD studies, the last 500–1000 bp of target gene, PfMaf1 Pf3D7_0416500, was cloned into pSLI-3HA-ddFKBP. Each sequence started with an in-frame stop codon but the stop codon at the end of the gene was removed. 50 µg of plasmid DNA was transfected into ring stage 3D7 *P. falciparum* parasites using the protocol described elsewhere

(*Hasenkamp et al., 2013*). Transfected parasites were selected with constant drug selection pressure of 2.56 nM WR99210 (Jacobus Pharmaceuticals) to obtain a cell line containing the episomal plasmid. A second drug selection using 400 µg/mL of G418 was done to select for integrants. Once parasites emerged, gDNA of each integration cell line was collected using a commercial kit (DNeasy Blood & Tissue Kit) and checked by PCR to show that integration occurred at the correct locus. Both genome and vector-specific primers for the 5' and 3' region were used so that the PCR product would cover the plasmid/genome junction. Vector primers used were the same as described (*Birnbaum et al., 2017*). Once proper size gel bands from PCR were seen, parasites were cloned by limiting dilution, and the targeted genomic locus was sequenced to confirm tag and FKBP integration.

### Flow cytometry

Two different pSLI-Maf1-FKBP parasite clones were tightly synchronized and diluted to 0.2% parasitemia (5% hematocrit) at ring stage. One cycle before, Shield-1 was removed from half of the culture. The growth curve was performed in a 96-well plate (200 µL culture per well) with three technical replicates of two biological replicates per condition per clone. At 0 hr, 24 hr, 48 hr, 72 hr, and 96 hr, 5 µL of the culture was stained in 95 µL of DPBS supplemented with 2× Sybr Green I (Ozyme; stock = 10,000×) for 30 min at room temperature, diluted 20-fold in DPBS (final volume = 200 µL), and the Sybr Green fluorescence measured in a Guava easyCyte Flow Cytometer (EMD Millipore). 30,000 events were counted in duplicate to establish an accurate parasitemia value for each culture. Data was analyzed using the InCyte software (EMD Millipore).

### Western blot analysis

Shield-1 was removed for 3 consecutive cycles to monitor the degradation of PfMaf1. Total protein extracts were prepared from trophozoite stages for control and −Shield-1 for 3 cycles. Additionally, membrane extracts were prepared for control and $MgCl_2$ supplementation from parasites isolated from plasmion enrichment. iRBCs were washed once with DPBS at 37°C and lysed with 0.075% saponin (Sigma S7900) in DPBS at 37°C. Parasites were washed once with DPBS, resuspended in 1 mL cytoplasmic lysis buffer (25 mM Tris-HCl pH 7.5, 10 mM NaCl, 1.5 mM $MgCl_2$, 1% IGEPAL CA-630, and 1× protease inhibitor cocktail ['PI', Roche 11836170001]) at 4°C, and incubated on ice for 30 min. Cells were further homogenized with a chilled glass douncer, and the cytoplasmic lysate was cleared with centrifugation (13,500×$g$, 10 min, 4°C). The pellet (containing the nuclei) was resuspended in 100 µL nuclear extraction buffer (25 mM Tris-HCl pH 7.5, 600 mM NaCl, 1.5 mM $MgCl_2$, 1% IGEPAL CA-630, PI) at 4°C and sonicated for 10 cycles with 30 s (on/off) intervals (5 min total sonication time) in a Diagenode Pico Bioruptor at 4°C. The nuclear lysate was cleared with centrifugation (13,500×$g$, 10 min, 4°C). Membrane extracts were prepared by resuspending parasite pellets in NETT buffer (50 mM Tris pH 8, 150 mM NaCl, 5 mM EDTA, 1% IGEPAL CA-630, PI) and incubated at 4°C for 10 min. Supernatant was removed after centrifugation (13,500×$g$, 10 min, 4°C) and the pellet was resuspended in Tris-saline buffer (50 mM Tris pH 8, 150 mM NaCl, 2% SDS, PI) and sonicated for 6 cycles with 30 s (on/off) intervals (3 min total sonication time) in a Diagenode Pico Bioruptor at 4°C. The membrane lysates were cleared with centrifugation (13,500×$g$, 10 min, 4°C). All protein samples were supplemented with NuPage Sample Buffer (Thermo Fisher NP0008) and NuPage Reducing Agent (Thermo Fisher NP0004) and denatured for 5 min at 95°C. Proteins were separated on a 4–15% TGX (Tris-Glycine eXtended) (Bio-Rad) and transferred to a PVDF membrane. The membrane was blocked for 1 hr with 5% milk in PBST (PBS, 0.1% Tween 20) at 25°C. HA-tagged proteins and histone H3 were detected with anti-HA (Abcam 9110, 1:1000 in 5% milk-PBST) and anti-H3 (Abcam ab1791, 1:1000 in 5% milk-PBST) primary antibodies, respectively, followed by donkey anti-rabbit secondary antibody conjugated to horseradish peroxidase ('HRP', Sigma GENA934, 1:5000 in 5% milk-PBST). Anti-ATS and anti-NapL were detected as previously described (*Nacer et al., 2015*; *Dawn et al., 2014*). Aldolase was detected with anti-aldolase-HRP (Abcam ab38905, 1:5000 in 5% milk-PBST). HRP signal was developed with SuperSignal West Pico chemiluminescent substrate (Thermo Fisher 34080) and imaged with a ChemiDoc XRS+ (Bio-Rad).

### Merozoite number per schizont analysis

Mature schizonts were assessed by microscopic analysis of Giemsa-stained smears and manually quantified using ImageJ software, as reported previously (*Marreiros et al., 2023*). 100 segmented

schizonts with clearly individualized merozoites containing a single hemozoin crystal were quantified per condition.

## Co-immunoprecipitation followed by mass spectrometry

PfMaf1-HA-ddFKBP tagged parasites (n=5 biological replicates) were synchronized. At 24 hpi, each culture ($1.5\times10^9$ parasites) was centrifuged and RBCs were lysed with six volumes of 0.15% saponin in DPBS for 5 min at 4°C. Parasites were centrifuged at 4000×$g$ for 5 min at 4°C, and the pellet was washed twice with DPBS at 4°C. Parasites were then cross-linked with 1% formaldehyde for 15 min at room temperature and quenched with 125 mM glycine for 5 min on ice. Cross-linked parasites were washed twice with DPBS and then resuspended in 900 µL of cytoplasmic lysis buffer (10 mM Tris-HCl pH 7.5, 1 mM EDTA, 0.65% IGEPAL CA-630, 10 mM NaCl) supplemented with protease inhibitors (Thermo Fisher 78440) at 4°C and incubated with rotation for 30 min at 4°C. Extracts were centrifuged for 10 min at 2000×$g$ at 4°C and the cleared cytoplasmic supernatant was removed and kept on ice. The nuclear pellet was resuspended in 900 µL nuclear lysis buffer (10 mM Tris-HCl pH 7.5, 500 mM NaCl, 1 mM EDTA, 1% sodium deoxycholate, 0.1% SDS, 1% IGEPAL CA-630, PI) at 4°C and transferred to 1.5 mL sonication tubes (300 µL per tube, DiagenodeC30010016). Samples were sonicated for 5 min (30 s on/off) in a Diagenode Pico Bioruptor at 4°C. Lysates were then centrifuged for 10 min at 13,500×$g$ at 4°C and supernatant was transferred to a fresh tube. Cytoplasmic fractions were mixed with 2:3 ratio of cytoplasmic dilution buffer (10 mM Tris-HCl pH 7.5, 150 mM NaCl, 0.5 mmM EDTA) and nuclear supernatants were mixed with 1:3 ratio of nuclear dilution buffer (10 mM Tris-HCl pH 7.5, 0.5 mmM EDTA). Cytoplasmic and nuclear fraction supernatants were incubated with 1 µg of α-HA antibody (Abcam 9110) and 25 µL Protein G Magnetic Dynabeads (Invitrogen), pre-incubated for a minimum of 2 hr and washed twice with dilution buffer, overnight with rotation at 4°C. The next day, the beads were collected on a magnet and the supernatant was removed. While on the magnetic stand, beads were washed twice with 500 µL wash buffer (10 mM Tris-HCl pH 7.5, 150 mM NaCl, 0.5 mM EDTA, 0.05% NP40), once with 25 mM $NH_4HCO_3$ (Sigma 09830) buffer, and then transferred to new tube. Finally, the beads were resuspended in 100 µL of 25 mM $NH_4HCO_3$ (Sigma 09830) and digested by adding 0.2 µg of trypsin-LysC (Promega) for 1 hr at 37°C. Samples were then loaded into custom-made C18 StageTips packed by stacking one AttractSPE disk (#SPE-Disks-Bio-C18-100.47.20 Affinisep) in a 200 µL micropipette tip for desalting. Peptides were eluted using a ratio of 40:60 $CH_3CN:H_2O + 0.1\%$ formic acid and vacuum concentrated to dryness with a SpeedVac apparatus. Peptides were reconstituted in 10 of injection buffer in 0.3% trifluoroacetic acid before liquid chromatography-tandem mass spectrometry (LC-MS/MS) analysis. Online LC was performed with an RSLCnano system (Ultimate 3000, Thermo Scientific) coupled to an Orbitrap Eclipse mass spectrometer (Thermo Scientific). Peptides were trapped on a 2 cm nanoviper Precolumn (i.d. 75 µm, C18 Acclaim PepMap 100, Thermo Scientific) at a flow rate of 3.0 µL/min in buffer A (2/98 MeCN/$H_2O$ in 0.1% formic acid) for 4 min to desalt and concentrate the samples. Separation was performed on a 50 cm nanoviper column (i.d. 75 µm, C18, Acclaim PepMap RSLC, 2 µm, 100 Å, Thermo Scientific) regulated to a temperature of 50°C with a linear gradient from 2% to 25% buffer B (100% MeCN in 0.1% formic acid) at a flow rate of 300 nL/min over 91 min. MS1 data were collected in the Orbitrap (120,000 resolution; maximum injection time 60 ms; AGC $4\times10^5$). Charges states between 2 and 7 were required for MS2 analysis, and a 60 s dynamic exclusion window was used. MS2 scan were performed in the ion trap in rapid mode with HCD fragmentation (isolation window 1.2 Da; NCE 30%; maximum injection time 60 ms; AGC 104).

For identification, the data were searched against the *Homo sapiens* (UP000005640_9606) and the *P. falciparum 3D7* (UP000001450_36329) UniProt databases using Sequest HT through proteome discoverer (version 2.4). Enzyme specificity was set to trypsin and a maximum of two miss cleavages sites were allowed. Oxidized methionine, Met-loss, Met-loss-Acetyl and N-terminal acetylation were set as variable modifications. Carbamidomethylation of cysteines were set as fixed modification. Maximum allowed mass deviation was set to 10 ppm for monoisotopic precursor ions and 0.6 Da for MS/MS peaks. The resulting files were further processed using myProMS version 3.9.3 (https://github.com/bioinfo-pf-curie/myproms; *Poullet, 2021*; *Poullet et al., 2007*). FDR calculation used Percolator (*The et al., 2016*) and was set to 1% at the peptide level for the whole study. The label-free quantification was performed by peptide Extracted Ion Chromatograms (XICs), reextracted across all conditions and computed with MassChroQ version 2.2.21 (*Valot et al., 2011*). For protein quantification,

XICs from proteotypic peptides shared between compared conditions (TopN matching) and missed cleavages were allowed. Median and scale normalization was applied on the total signal to correct the XICs for each biological replicate (n=5 in each condition). To estimate the significance of the change in protein abundance, a linear model (adjusted on peptides and biological replicates) was performed, and p-values were adjusted using the Benjamini-Hochberg FDR procedure. Proteins with at least two distinct peptides in three replicates of a same state, a twofold enrichment, and an adjusted p-value≤0.05 were considered significantly enriched in sample comparisons. Proteins unique to a condition were also considered if they matched the peptides criteria.

### IF assay

pSLI-Maf1-FKBP control, +$MgCl_2$ parasites were used with rat anti-HA (Roche 3F10) antibodies. 10 μL of iRBCs were washed with PBS and fixed with for 30 min in 0.0075% glutaraldehyde/4% PFA/DPBS. After DPBS washing, parasites were permeabilized with 0.1% Triton X-100/PBS for 10–15 min before quenching free aldehyde groups with $NH_4Cl$ solution for 10 min. Next, parasites were blocked with 3% BSA-DPBS for 30 min. Primary antibody incubation (1:500 dilution) lasted for 1 hr before three washes with PBS, and secondary antibody incubation for 30–60 min, Alexa Fluor 488-conjugated anti-mouse IgG (Invitrogen) diluted 1:2000 in 4% BSA-DPBS. After three final washes in DPBS, cells were mounted in Vectashield containing DAPI for nuclear staining. Images were captured using a Delta Vision Elite microscope (GE Healthcare). Image overlays were generated using Fiji (*Schindelin et al., 2012*).

### Co-immunoprecipitation followed by western blot

PfMaf1-HA-ddFKBP tagged parasites were synchronized and split into control and $MgCl_2$ supplementation. At 18 hpi, each culture (1.5×10$^9$ parasites) was centrifuged and RBCs were lysed with six volumes of 0.15% saponin in DPBS for 5 min at 4°C. Parasites were centrifuged at 4000×$g$ for 5 min at 4°C, and the pellet was washed twice with DPBS at 4°C. The parasite pellets were then resuspended in 900 μL of cytoplasmic lysis buffer (10 mM Tris-HCl pH 7.5, 1 mM EDTA, 0.65% IGEPAL CA-630, 10 mM NaCl) supplemented with protease inhibitors (Thermo Fisher 78440) at 4°C and incubated with rotation for 30 min at 4°C. Extracts were centrifuged for 10 min at 2000×$g$ at 4°C and the cleared cytoplasmic supernatant was removed and kept on ice. The nuclear pellet was resuspended in 200 μL nuclear lysis buffer (10 mM Tris-HCl pH 7.5, 500 mM NaCl, 1 mM EDTA, 1% sodium deoxycholate, 0.1% SDS, 1% IGEPAL CA-630, PI) at 4°C and transferred to 1.5 mL sonication tubes (300 μL per tube, DiagenodeC30010016). Samples were sonicated for 5 min (30 s on/off) in a Diagenode Pico Bioruptor at 4°C. Lysates were then centrifuged for 10 min at 13,500×$g$ at 4°C and supernatant was transferred to a fresh tube. Cytoplasmic fractions were mixed with 2:3 ratio of cytoplasmic dilution buffer (10 mM Tris-HCl pH 7.5, 150 mM NaCl, 0.5 mmM EDTA) and nuclear supernatants were mixed with 1:3 ratio of nuclear dilution buffer (10 mM Tris-HCl pH 7.5, 0.5 mmM EDTA). Cytoplasmic and nuclear fraction supernatants were incubated with 1 μg of α-HA antibody (Abcam 9110) and 25 μL Protein G Magnetic Dynabeads (Invitrogen), pre-incubated for a minimum of 2 hr and washed twice with dilution buffer, overnight with rotation at 4°C. The next day, the beads were collected on a magnet and the supernatant was removed. While on the magnetic stand, beads were washed twice with 500 μL wash buffer (10 mM Tris-HCl pH 7.5, 150 mM NaCl, 0.5 mM EDTA, 0.05% NP40), once with 25 mM $NH_4HCO_3$ (Sigma 09830) buffer, and then transferred to new tube. Finally, the beads were resuspended in 20 μL 2× SDS sample buffer and reducing agent before 5 min at 95°C. The beads were separated with magnet and the supernatant was transferred to a new tube. Western blot protocol was followed as mentioned earlier.

### Static cytoadhesion binding assay

Mature stage iRBCs for 3D7 control and supplementation of $MgCl_2$ ([3 mM] total concentration) for 1 cycle were used for cytoadhesion binding assays. Receptors diluted in PBS (10 μg/mL target CD36 or negative control 1% BSA) were incubated overnight at 4°C in labeled Petri dishes. After the dishes were blocked for 30 min at 37°C with 1% BSA/PBS, parasitemia was measured with a Guava easyCyte Flow Cytometer (EMD Millipore) after trophozoites and schizonts iRBCs were isolated by plasmagel (Plasmion, Fresenius Kabi) enrichment. Adjusted amounts of iRBCs were resuspended in binding medium (RPMI 1640 powder [Gibco 51800: W/L-Glutamine W/O $NaHCO_3$] with HEPES pH 7) for a concentration of 2.2×10$^8$ iRBCs/mL. iRBCs/binding medium was added to each Petri dish and

incubated for 2 hr at 37°C. After, unbound cells were removed and the dishes were washed five times with binding medium by carefully tilting from side to side. Adherent iRBCs were counted using 40× lens with a Nikon ECLIPSE TE200 in 10 randomly selected fields (each with 0.2 μm²) before our fixation with 2% glutaraldehyde (G5882; Sigma)/PBS. After fixation, Giemsa staining was done to confirm percentage of bound iRBCs. A total of three biological replicates were performed and the results were expressed as the number of bound iRBCs per 0.2 mm² of target receptor monolayer.

## Statistical analysis

All statistical analyses were performed using GraphPad Prism version 9.1.0 (216) for Mac. To test for a normal distribution of the data, the Shapiro-Wilk normality test was used. To test for significance between two groups, a two-sided independent-samples t-test was used. GO enrichments were calculated using the build-in tool at https://plasmoDB.org.

## Estimation of cell cycle progression

RNA-seq-based cell cycle progression for control and $MgCl_2$ supplementation was estimated in R by comparing the normalized expression values (i.e. RPKM, reads per kilobase per exon per 1 million mapped reads) of each sample to the microarray data from *Bozdech et al., 2003*; Data *Bozdech et al., 2003* using the statistical model as in *Lemieux et al., 2009*.

## Acknowledgements

We thank L Mancio Silva, E Real, and J Bryant for critical reading this manuscript. This work was supported by the Laboratoire d'Excellence (LabEx) ParaFrap [ANR-11-LABX-0024], ERC AdG PlasmoSilencing (670301), ANR-JC (18-CE15-0009-01). We thank Camille Cohen and Marta Miera Maluenda for technical assistance with this project. With financial support from 'la Région Île-de-France' (N°EX061034) and ITMO Cancer of Aviesan and INCa on funds administered by Inserm (N°21CQ016-00) for MS analysis.

## Additional information

### Funding

| Funder | Grant reference number | Author |
|---|---|---|
| Agence Nationale de la Recherche | ANR-11-LABX-0024 | Artur Scherf |
| Agence Nationale de la Recherche | 18-CE15-0009-01 | Artur Scherf |
| European Research Council | AdG PlasmoSilencing 670301 | Artur Scherf |

The funders had no role in study design, data collection and interpretation, or the decision to submit the work for publication.

### Author contributions

Gretchen Diffendall, Conceptualization, Validation, Investigation, Visualization, Writing - original draft, Writing - review and editing; Aurelie Claes, Anna Barcons-Simon, Prince Nyarko, Investigation; Florent Dingli, Data curation, Formal analysis; Miguel M Santos, Formal analysis, Aided in investigation under supervision; Damarys Loew, Supervision; Antoine Claessens, Supervision, Funding acquisition; Artur Scherf, Supervision, Funding acquisition, Writing - original draft, Writing - review and editing

### Author ORCIDs

Gretchen Diffendall http://orcid.org/0000-0002-0783-517X
Miguel M Santos https://orcid.org/0000-0002-5594-2682
Damarys Loew http://orcid.org/0000-0002-9111-8842
Antoine Claessens http://orcid.org/0000-0002-4277-0914
Artur Scherf http://orcid.org/0000-0003-2411-3328

## Ethics

Human subjects: Venous blood draw of different infected individuals in The Gambia during the dry and wet seasons was collected as previously described in (Collins et al, 2022; Fogang et al, 2023). The study protocol was reviewed and approved by the Gambia Government/MRC Joint Ethics Committee (SCC 1476, SCC 1318, L2015.50) and by the London School of Hygiene & Tropical Medicine ethics committee (Ref 10982). The field studies were also approved by local administrative representatives, the village chiefs. Written informed consent was obtained from participants over 18 years old and from parents/guardians for participants under 18 years. Written assent was obtained from all individuals aged 12-17 years.

Reviewer #1 (Public Review): https://doi.org/10.7554/eLife.95879.3.sa1
Reviewer #2 (Public Review): https://doi.org/10.7554/eLife.95879.3.sa2
Reviewer #3 (Public Review): https://doi.org/10.7554/eLife.95879.3.sa3
Author response https://doi.org/10.7554/eLife.95879.3.sa4

# Additional files

## Supplementary files

• Supplementary file 1. Quantitative PCR (qPCR) analysis primer pairs. Primer pairs used are listed including the forward and reverse sequences.

• Supplementary file 2. Samples used for plasma magnesium concentration assay. Age and sex are indicated for each sample as well as the month each sample was from. Green highlight shows magnesium concentration and purple highlight shows which samples were also used for reverse transcription-quantitative PCR (RT-qPCR) analysis in *Figure 1*.

• MDAR checklist

## Data availability

Figure 1-source data 1, Figure 2-source data 1, Figure 3-source data 1 contain the numerical data used to generate the figures. The data generation in this study is available in the following database: The mass spectrometry proteomics data have been deposited to the ProteomeXchange Consortium via the PRIDE (*Perez-Riverol et al., 2022*) partner repository with the dataset identifier PXD040576.

The following dataset was generated:

| Author(s) | Year | Dataset title | Dataset URL | Database and Identifier |
|-----------|------|---------------|-------------|-------------------------|
| Scherf A | 2024 | Environmental inhibition of RNA Pol III reduces malaria parasite virulence (during asymptomatic infection) | https://www.ebi.ac.uk/pride/archive/projects/PXD040576 | PRIDE, PXD040576 |

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
