## [Editor Report · eLife assessment]

This **important** study links the activity of polymerase III to the regulation of virulence gene expression in the deadliest malaria parasite, *Plasmodium falciparum*. It identifies Maf1 as a Pol III inhibitor that enables the parasite to respond to external stimuli such as magnesium chloride plasma levels by downregulating Pol III-transcribed ruf6 genes and subsequently regulated var genes. While the evidence presented is generally **convincing**, some of the results are **incomplete**, and the mechanistic link between external signals and Maf1 activation remains unknown.

---

## [Referee Report · Reviewer #1 (Public Review)]

Summary:

Asymptomatic malaria infections are frequent during the dry season and have been associated with lower cytoadherence of *P. falciparum* parasites and lower expression of variant surface antigens. The mechanisms underlying parasite adaptation during the low transmission season remain poorly understood. The authors previously established that members of the non-coding RNA RUF6 gene family, transcribed by RNA pol III, are required for expression of the main variant surface antigens in *P. falciparum*, PfEMP1, which drive parasite cytoadherence and pathogenicity. In this study, the authors investigated the contribution of RNA pol III transcription in the regulation of PfEMP1 expression in different clinical states, either symptomatic malaria cases during the wet season or asymptomatic infections during the dry season.

By reanalyzing RNAseq data from a previous study in Mali, complemented with RT-qPCR on new samples collected in The Gambia, the authors first report the down-regulation of RNA pol III genes (tRNAs, RUF6) in *P. falciparum* isolates collected from asymptomatic individuals during the dry season, as compared to isolates from symptomatic (wet season) individuals. They also confirm the down-regulation of var (DBLalpha) gene expression in asymptomatic infection as compared to symptomatic malaria. Plasma analysis in the two groups in the Gambian study reveals higher Magnesium levels in dry season as compared to wet season samples, pointing at a possible role of external factors. The authors tested the effect of MgCl2 supplementation on cultured parasites, as well as three other stimuli (temperature, low glucose, Ile deprivation), and show that Ile deprivation and MgCl2 both induce down-regulation of RNA pol III transcription but not pol I or pol II (except the active var gene). Using RNAseq, they show that MgCl2 supplementation predominantly inhibits RNA pol III-transcribed genes, including the entire RUF6 family. Conditional depletion of Maf1 leads to the up-regulation of RNA pol III gene transcription, confirming that Maf1 is a RNA pol III inhibitor in *P. falciparum*, as described in other organisms. Quantitative mass spectrometry shows that Maf1 interacts with RNA pol III complex in the nucleus, and with distinct proteins including two phosphatases in the cytoplasm. Using the Maf1 cKD parasites, the authors document that down-regulation of RNA pol III by MgCl2 is dependent on Maf1. Finally, they show that MgCl2 results in decreased cytoadherence of infected erythrocytes, associated with reduced PfEMP1 expression.

Strengths:

-The work is very well performed and presented.

-The study uncovers a novel regulatory mechanism relying on RNA pol III-dependent regulation of variant surface antigens in response to external signals, which could contribute to parasite adaptation during the low transmission season.

-Potential regulators of Maf1 were identified by mass spectrometry, including phosphatases, paving the way for future mechanistic studies.

Weaknesses:

-The signaling pathway upstream of Maf1 remains unknown. In eukaryotes, Maf1 is a negative regulator of RNA pol III and is regulated by external signals via the TORC pathway. Since TORC components are absent in the apicomplexan lineage, one central question that remains open is how Maf1 is regulated in *P. falciparum*. Magnesium is probably not the sole stimulus involved, as suggested by the observation that Ile deprivation also down-regulates RNA pol III activity.

-The study does not address why MgCl2 levels vary depending on the clinical state. It is unclear whether plasma magnesium is increased during asymptomatic malaria or decreased during symptomatic infection, as the study does not include control groups with non-infected individuals. Along the same line, MgCl2 supplementation in parasite cultures was done at 3mM, which is higher than the highest concentrations observed in clinical samples.

-Although the study provides biochemical evidence of Maf1 accumulation in the parasite nuclear fraction upon magnesium addition, this is not fully supported by the immunofluorescence experiments.

---

## [Referee Report · Reviewer #2 (Public Review)]

The study by Diffendall et al. set out to establish a link between the activity of RNA polymerase III (Pol III) and its inhibitor Maf1 and the virulence of Plasmodium falciparum in vivo. Having previously found that knockdown of the ncRNA ruf6 gene family reduces var gene expression in vitro, they now present experimental evidence for the regulation of ruf6 and subsequently, var gene expression by Pol III using a commercially available inhibitor. They confirm their findings with samples from a previously published Gambian cohort study using asymptomatic dry season and mildly symptomatic wet season samples, showing that higher levels of Pol III-dependent transcripts and var transcripts as well as lower MgCl2 plasma concentrations are present in wet season samples. From this, they hypothesize that the external stimuli heat, reduced glucose and essential amino acid supply, and increased MgCl2 levels are sensed by the parasite through the only known Pol III inhibitor Maf1 and result in lower Pol III activity and fewer ruf6 transcripts, which in turn reduces var gene expression, leading to reduced cytoadherence and virulence of *P. falciparum*. In their in vitro experiments they focus on investigating higher MgCl2 levels and their impact on Pol III and Maf1 activity as well as var gene expression and parasites adherence to purified CD36, thereby successfully confirming their hypothesis for MgCl2. Nicely, MgCl2-induced down-regulation of Pol III activity was shown to be dependent on Maf1 using a knock-down cell line. Additionally, they show that the Maf1-KD cell line displays a slower growth rate with fewer merozoites per schizonts and Maf1 interacts with RNA pol III subunits and some kinases/phosphatases.

Comments on latest version:

It is understandable that the RNA samples from the Gambian cohort were limited, but for all in vitro analyses a larger panel of qPCR primers or RNAseq would have been feasible. I also understand the rationale for using the general var primer pair (DBLa) for field isolates, but since the authors were working with a clonal parasite line (3D7) in vitro, qPCR with specific 3D7/NF54 primer pairs or RNAseq, which would also allow inferences about ruf6 regulation of specific (neighboring?) var genes and other Pol III-regulated genes, would have been a far better option.

As far as I could see from the resubmitted manuscript, the authors did not correct the statistical analyses. For example, they continue to apply a t-test to fold-change values (which must be transformed to log2), many t-test based analyses rely on only 2-3 replicates (a non-parametric test would be more appropriate), they have not corrected for multiple testing, and it is unclear how the authors handle technical and biological replicates in their plots. Therefore, I still suspect that more appropriate statistical analyses might have an impact on the significance of their results.

I agree that CD36 binding is associated with mild malaria, but since the authors only make a link between Pol III and CD36 binding in vitro, I think it is an overstatement to claim something like "Our study reveals a regulatory mechanism in *P. falciparum* involving RNA Polymerase III, which plays a pivotal role in the parasite's virulence."

Finally, if the authors have checked all the relevant literature on MgCl2, it should be easy for them to give a brief explanation why they included only one study and ignored all the other contradictory results.

---

## [Referee Report · Reviewer #3 (Public Review)]

Summary:

This work describes a new pathway by which malaria parasites, *P. falciparum*, may regulate their growth and virulence (i.e. their expression of virulence-linked cytoadhesins). This is a topic of considerable interest in the field - does this important parasite sense factor(s) in its host bloodstream and regulate itself accordingly? Several fragments of evidence have come out on this topic in the past decade, showing, for example, reduced parasite growth under calorie restriction (in mice); parasite dormancy in response to amino acid starvation (in culture and in mice), and also reduced virulence in dry-season, low-parasitaemia infections in humans. The molecular mechanisms that may underlie this interesting biology remain only poorly understood.

Here, the authors show that dry-season *P. falciparum* parasites have reduced expression of Pol3-transcribed tRNAs and ncRNAs that positively regulate virulence gene expression. They link the level of Pol3 activity to PfMaf1, a remnant of the largely-absent nutrient-sensing TOR pathway in this parasite. They propose that in the dry season, human hosts may be calorie-restricted, leading to Maf1 moving to the nucleus and suppressing Pol3, thus downregulated growth and virulence of parasites. The evidence is intriguing and the idea is conceptually elegant.

Strengths:

The use of dry/wet-season field samples from The Gambia is a strength, showing potential real-world relevance. The generation of an inducible knockdown of Maf1 in lab-cultured parasites is also a strength, allowing this pathway to be studied somewhat in isolation.

Weaknesses:

(1) The signals upstream of Maf1 remain rather a black box. 4 are tested - heatshock and low-glucose, which seem to suppress ALL transcription; low-Isoleucine and high magnesium, which suppress Pol3. Therefore the authors use Mg supplementation throughout as a 'starvation type' stimulus. They do not discuss why they didn't use amino acid limitation, which could be more easily rationalised physiologically. It may for experimental simplicity (no need for dropout media) but this should be discussed, and ideally sample experiments with low-IsoLeu should be done too, to see if the responses (e.g. cytoadhesion) are all the same.

(2) The proteomics, conducted to seek partners of Maf1, is probably the weakest part. From Fig S4 it is clear that the proteins highlighted in the text are highly selected (as ones that might be relevant, e.g. phosphatases), but many others are more enriched. It would be good to see (a) the top hits from the whole list provided as a short table within the main proteomics figure, along with the GO terms that actually came top in enrichment; (b) the whole list provided as a supp. spreadsheet for easy re-analysis, rather than a PDF which cannot be easily re-used.

(3) Fig 3 shows the Maf1-low line has very poor growth after only 5 days but it is stated that no dead parasites are seen even after 8 cycles and the merozoites number is down only ~18 to 15... is this too small to account for such poor growth (~5-fold reduced in a single cycle, day 3-5)? It would additionally be interesting to see a cell-cycle length assessment and invasion assay, to see of Maf1-low parasite have further defects in growth.

Other weaknesses, which are more restricted but were not addressed in revision, are highlighted below:

Fig S1B - The downregulation of RNAPol3 transcripts caused by a commercial Pol3 inhibitor is pretty weak - mostly non-significant. The authors might comment on why they think this is, when interfering with PfMaf1 evidently has a greater effect.

Fig 2D: the legend states ' Expressed transcripts from three replicates between control and addition of MgCl2 that are significantly up-regulated are highlighted in red while significantly down-regulated RNA Pol III genes are highlighted in blue (FDR corrected p-value of <0.05) and a FC {greater than or equal to}{plus minus} 1.95 with examples listed as text'. This isn't very clear. The authors could clarify whether they took ALL (Pol3 or not) upregulated genes to show in red, but only putative Pol3-regulated genes to show in blue? If so, why? Or did they take all significantly downregulated genes, and found they were all annotated as pol3 transcribed? (I cannot see any dots that are not blue. If there are some, a clearer figure is needed?)

Line 227: 'PfMaf1 levels were shown to decrease by approximately 57% in total extracts after one cycle' - the provenance of this very precise percentage isn't clear (it does not appear on the figure). Is it densitometry of a western blot? And if so, is it an average of the 3 replicates that are stated in the legend (but not shown), or from the single example blot shown in Figure 3?

Fig 4A: the western blot, as shown, lacks controls, both for loading and for completeness of cyto/nuclear fractionation. To avoid confusion, these should be shown in the main figure, as is standard in the field, rather than separately in a supp figure. Ideally, 3 repeats should be done, with densitiometry quantification.

---

## [Author Response]

The following is the authors’ response to the original reviews.

**Public Reviews:**

Here we address the major points raised by the reviewers.

**Reviewer #1 (Public Review):**
Weaknesses:The signaling pathway upstream of Maf1 remains unknown. In eukaryotes, Maf1 is a negative regulator of RNA pol III and is regulated by external signals via the TORC pathway. Since TORC components are absent in the apicomplexan lineage, one central question that remains open is how Maf1 is regulated in *P. falciparum*. Magnesium is probably not the sole stimulus involved, as suggested by the observation that Ile deprivation also down-regulates RNA pol III activity.

We agree that there is still much to uncover relating to the PfMaf1 signaling pathway. While we still do not know each component, we have been able to link external factors (of course not limited to only magnesium) to the increased nuclear occupancy of PfMaf1. Other protein interactors that potentially regulate PfMaf1, while not confirmed, have been identified in plasma sample as candidates for future experiments to validate their potential involvement of RNA Pol III inhibition.

The study does not address why MgCl2 levels vary depending on the clinical state. It is unclear whether plasma magnesium is increased during asymptomatic malaria or decreased during symptomatic infection, as the study does not include control groups with non-infected individuals. Along the same line, MgCl2 supplementation in parasite cultures was done at 3mM, which is higher than the highest concentrations observed in clinical samples.

This reviewer raised a valid point. The plasma magnesium levels for the wet symptomatic samples (averaging [0.79mM]) were within the normal range of a healthy individual (between [0.75-0.95mM]) while the dry asymptomatic levels were above the normal range (averaging [1.13mM]). Ideally, we would have liked to have control uninfected plasma samples from individuals from The Gambia. Unfortunately, field studies and human volunteer studies do not always have all the ideal controls that in vitro studies have. We recognize that [3mM] is higher than the normal range for magnesium levels, which is why we included a revised Supplementary Figure 3A. This figure shows that magnesium concentrations as low as [1mM] (similar to the levels found in dry asymptomatic samples) reduced the expression of RNA Pol III-transcribed genes.

Although the study provides biochemical evidence of Maf1 accumulation in the parasite nuclear fraction upon magnesium addition, this is not fully supported by the immunofluorescence experiments.

We agree that the resolution of IFA images does not allow to support the WB data. We believe that the importance of the IFA Supplementary Figure is to show that PfMaf1 clusters together in foci, which has not been previously reported.

**Reviewer #2 (Public Review):**
Weaknesses:However, most analyses are rather preliminary as only very few (3-5) candidate genes are analyzed by qPCR instead of carrying out comprehensive analyses with a large qPCR panel or RNA-seq experiments with GO term analyses. Data presentation lacks clarity, the number of biological replicates is rather low and the statistical analyses need to be largely revised. Although the in vivo data from wet (mildly symptomatic) and dry (asymptomatic) season parasites with different expression levels of Pol III-regulated genes, var genes, and MgCl2 are interesting, the link between the in vitro data and the in vivo virulence of *P. falciparum*, which is made in many sections of the manuscript, should be toned down. Especially since (i) the only endothelial receptor studied is CD36, which is associated with parasite binding during mild malaria, and (ii) several studies provide contradictory data on MgCl2 levels during malaria and in different disease states, which is not further discussed, but the authors mainly focused on this external stimulus in their experiments.

We agree that, ideally, we would have liked to do full RNA-seq on The Gambia samples. However, that was out of the scope of this project. The RNA samples were limited which is why we did not use more primers. We believe that an appropriate number of replicates was done for the experiments. The wet symptomatic samples from this study were from mildly symptomatic individuals, as stated in the manuscript. Therefore, CD36 was a relevant receptor to use for our studies.

We agree that the published studies about magnesium levels in infected individuals are not always consistent. What these studies do not consider is the time of year, whether the infection occurred during the dry or wet season. These studies were also done in different regions of the world using different technologies. For this reason, we only highlight the observed difference observed in our field study data from The Gambia.

**Reviewer #3 (Public Review):**
Weaknesses:(1) The signals upstream of Maf1 remain rather a black box. 4 are tested - heat shock and low-glucose, which seem to suppress ALL transcription; low-Isoleucine and high magnesium, which suppress Pol3. Therefore the authors use Mg supplementation throughout as a 'starvation type' stimulus. They do not discuss why they didn't use amino acid limitation, which could be more easily rationalised physiologically. It may be for experimental simplicity (no need for dropout media) but this should be discussed, and ideally, sample experiments with low-IsoLeu should be done too, to see if the responses (e.g. cytoadhesion) are all the same.

We agree that deprivation of isoleucine would have been another experimental assay for our study, but it also would not have been as novel as magnesium. While understanding the exact mechanism or involvement of magnesium as a stress condition was not the scope of this manuscript, we believe that our data will be valuable into demonstrating that external stimuli act on *P. falciparum* virulence gene expression via RNA Pol III inhibition. Since we also had plasma level data for magnesium, and not isoleucine, we believed it made for a better external factor to use for our in vitro studies.

(2) The proteomics, conducted to seek partners of Maf1, is probably the weakest part. From Figure S3: the proteins highlighted in the text are clearly highly selected (as ones that might be relevant, e.g. phosphatases), but many others are more enriched. It would be good to see the whole list, and which GO terms actually came top in enrichment.

We apologize if the reviewer did not see the attached supplementary Co-IP MS data. The file includes all proteins found in each sample as well as GO term analysis. For the purpose of this work, we highlight proteins potentially involved in the canonical role of Maf1 that have been shown in model organisms to reversibly inhibit RNA Pol III (phosphatases, RNA Pol III subunits).

(3) Figure 3 shows the Maf1-low line has very poor growth after only 5 days but it is stated that no dead parasites are seen even after 8 cycles and the merozoites number is down only ~18 to 15... is this too small to account for such poor growth (~5-fold reduced in a single cycle, day 3-5)? It would additionally be interesting to see a cell-cycle length assessment and invasion assay, to see if Maf1-low parasites have further defects in growth.

We agree with the reviewer that the observed reduced merozoite numbers may not the only cause of the reduced growth rate. Other factors in the PfMaf1 knock-down line may contribute to the observed poor growth.